# The Association between Post-Traumatic Stress Disorder, 5HTTLPR, and the Role of Ethnicity: A Meta-Analysis

**DOI:** 10.3390/genes15101270

**Published:** 2024-09-27

**Authors:** Marta Landoni, Sonia Di Tella, Giulia Ciuffo, Chiara Ionio

**Affiliations:** 1Developmental and Educational Dynamics Research Center (CRIdee), Università Cattolica del Sacro Cuore, 20123 Milan, Italy; giulia.ciuffo@unicatt.it (G.C.); chiara.ionio@unicatt.it (C.I.); 2Department of Psychology, Università Cattolica del Sacro Cuore, 20123 Milan, Italy; sonia.ditella@unicatt.it

**Keywords:** post-traumatic stress disorder, PTSD, 5HTTLPR, meta-analysis, systematic review, pregnancy, ethnicity

## Abstract

Background/Objectives: The current meta-analysis looks at the effect of ethnicity on the connection between 5-HTTLPR SNPs and PTSD patients in all published genetic association studies. Techniques: In accordance with PRISMA principles, the literature was searched in PubMed, Scopus, and ScienceDirect. A consistent method was followed by two reviewers who independently chose publications for inclusion and extracted data. Using a random-effects model, a meta-analysis of the biallelic and triallelic studies was conducted in order to determine the pooled OR and the associated 95% CI. The impact estimates were corrected for minor study effects, including publication bias, using the trim-and-fill approach. Findings: After 17 studies were deemed eligible for inclusion, the overall sample size was 8838 controls and 2586 PTSD patients, as opposed to 627 and 3524 in the triallelic meta-analysis. The results of our meta-analysis and comprehensive review do not point to a direct main effect of the 5-HTTLPR polymorphisms on PTSD. Nonetheless, preliminary data suggest that ethnicity influences the association between 5-HTTLPR and PTSD. Conclusions: According to our findings, ethnicity—especially African ethnicity—has a major influence on the relationship between 5-HTTLPR and PTSD and needs to be taken into account as a crucial moderating factor in further studies.

## 1. Introduction

A tiny percentage of people develop post-traumatic stress disorder (PTSD), a chronic and usually disabling mental illness, after one or more stressful events over time. Re-experiencing (flashbacks and nightmares), avoidance of triggers related to the trauma, hyperarousal, and negative thoughts and mood swings are the four primary categories of symptoms associated with post-traumatic stress disorder (PTSD) according to the DSM V. PTSD has often been considered a serious public health problem since it was included in the third version of the Diagnostic and Statistical Manual of Mental Disorders (DSM-III).

Furthermore, research has shown that the lifetime incidence of PTSD can vary from 1.9% [1] to 8.8% [2], tripling in communities affected by conflict [3] and rising to almost 23% [4] in earthquake survivors. According to several studies [5,6,7,8,9,10], PTSD is the most typical psychopathological reaction to an earthquake event. A number of factors have been associated with the risk of post-traumatic stress disorder (PTSD) and other psychological disorders following an earthquake [11]. These include the experiences had during and after the event, the characteristics of the affected individuals and their communities before the disaster, and the interactions between the two main categories of risk factors [11,12,13]. Although most people report having experienced one or more traumatic events in their lives [14], only a tiny proportion of these people develop post-traumatic stress disorder, suggesting that trauma exposure is a necessary but not sufficient condition for PTSD [14]. It is true that PTSD is a complex, multifaceted mental illness [11,15]. There is general agreement that heredity plays a role in its etiology, while environmental variables play a significant role in its development [4,11,16]. According to twin studies, one-third of the variation in PTSD can be explained by genetic variables [17]. Given our current understanding of the neurobiology of the disorder, numerous potential genes have been explored [18]. Due to variations in its promoter region (*5-HTTLPR*), the human serotonin transporter gene (5-hydroxytryptamine transporter, *5-HTT*) (*SLC6A4*) is one of the most frequently studied genetic variants [19,20]. *SLC6A4* is located on chromosome 17q11.1–q12 and encodes *5-HTT*. A long (L) and a short (S) allele result from a 43 bp insertion or deletion in the *5-HTT* promoter region (*5-HTTLPR*) with repeat elements from 6 to 8.

The more common long allele (L) in this polymorphism has higher transcriptional efficiency than the less common short allele (S) [19]. An experimental model has also been investigated [21], as a third functional allele, LG, has been described [11,22], which has comparable expression to the S allele. Therefore, *5-HTTLPR* can be considered as a triallelic locus with alleles LG, LA, and S, allowing functional reclassification or genotype sorting according to allele expression level [11,23,24]. The increasing evidence for the function of this gene in the regulation of stress sensitivity and susceptibility to psychopathology is an important factor in its significance. Numerous studies have studied possible associations between the PTSD environment and *5HTTLPR* [25,26,27,28,29,30,31,32,33], with both the S and L alleles producing notable results. Some studies have found significant interactions with the S allele, although this conclusion has not been verified by other researchers [34,35]. Three meta-analyses that analyzed studies on the association between *5-HTTLPR* and PTSD concluded that a correlation could not be established [34,36,37,38].

One of these studies focussed mainly on the relationship between stress, PTSD, and *5HTTLPR.* Nevertheless, a number of other studies have analyzed the relationship between *5HTTLPR* and PTSD in recent years. In light of this, the aim of this study is to clarify the role of the serotonin transporter gene polymorphism (*5-HTTLPR*) in the development of post-traumatic stress disorder (PTSD), a multifactorial disorder characterized by both genetic and environmental influences. Previous research examining the association between *5-HTTLPR* and PTSD has yielded conflicting results, emphasizing the need for further investigation. This meta-analysis attempts to resolve these contradictions by reassessing the relationship between *5-HTTLPR* and PTSD and comparing the traditional biallelic model (L- and S-alleles) with the refined triallelic model (LA-, LG-, and S-alleles) to more accurately capture the effects of the gene. We will also investigate how ethnicity influences this association, as allele frequencies vary considerably in different populations. For example, the long allele (L) is more common in Europeans (57%; 95% CI: 49.9–61.8%) than in Asians (27%; 95% CI: 23.9–32.9%), while the distribution of trialleles differs between African American and European-American populations. Understanding these genetic variations is critical, as individual susceptibility to PTSD may be closely linked to the presence or absence of specific polymorphisms that increase risk. By summarizing these findings, this study aims to deepen our understanding of the genetic basis of PTSD and contribute to the development of personalized, targeted interventions for high-risk populations, particularly those who have been exposed to trauma.

## 2. Materials and Methods

### 2.1. Search Strategy

We identified candidate studies according to the Preferred Reporting Items for Systematic Reviews and Meta-Analyses Guidelines (PRISMA) by searching multiple databases (Pubmed, Scopus, and Science of Direct) with the string: “Posttraumatic Stress Disorder OR PTSD AND 5HTTLPR OR 5HHT OR serotonin transporter gene”. In addition, we performed manual searches on Google Scholar. References to previous meta-analyses and review articles were also reviewed to identify eligible publications. To identify potentially relevant studies, two reviewers (M.L. and C.I.) conducted independent searches.

### 2.2. Inclusion and Exclusion Criteria

Included studies met the following criteria: (a) original research; (b) publication in indexed journals; (c) language; (d) population exposed to trauma and experiencing PTSD; (e) genotypic frequency registration for 5-HTTLPR; (f) case–control association studies that investigated the relationship between the diagnosis of 5-HTTLPR polymorphism and PTSD or association studies.

### 2.3. Exclusion Criteria

Books, chapters of books, qualitative studies, letters to the editor, commentaries, population studies with only healthy subjects, case reports, pilot, feasibility, and protocol studies, reviews, case studies, family-based designs, and other studies describing genetic effects on other anxiety- or depression-related phenotypes such as anxiety, depression or various personality traits were not included.

Non-English language studies were also disqualified.

Two separate, double-blinded reviewers found the articles using the search term and eliminated duplicates. Next, the titles and abstracts of the articles were checked. The two reviewers then went through the selected articles together to ensure that the research was suitable for review. Any discrepancies were clarified during a discussion.

### 2.4. Statistical Analysis

Using a biallelic model (Biallelic Frequency Model, BFM) and a triallelic model (Triallelic Frequency Model, TFM), statistical analyses were used to investigate the association between the frequency of *5-HTLLPR* alleles and PTSD.

As certain variation between the studies was to be expected, models with random effects were included in the statistical analysis. It is therefore assumed that the results of the different studies actually differ from each other and that the heterogeneity between the studies is taken into account in the calculations. A pooled OR and its 95% confidence interval were calculated for each meta-analysis. In addition, the statistical significance of the pooled OR was assessed using the Z-test. The I2 statistic was used to calculate the proportion of true variance in the total observed variation along with a 95% confidence interval (CI). 25%, 50%, and 75% of I2 values indicate a slight, moderate, and severe deviation from the true effect size, respectively [39].

To investigate possible publication bias, funnel plots were used to find asymmetries that would indicate a small study effect. Finally, after taking publication bias into account, the estimated number of missing studies was determined using the trim-and-fill approach. This resulted in an estimate of the magnitude of the effect. The trim-and-fill method was used to provide effect estimates that were corrected for tiny study effects, such as publication bias when asymmetry of the funnel plot was detected. This method recalculates the overall effect that would result if these studies were included by estimating the number and results of the missing (unreported) studies from the existing data. Each two-sided statistical test has a significance level of 5% (α = 0.05). Post hoc power (ω^2^) was calculated for all meta-analytic models adopting resources found in the *metafor* package in RStudio statistical software (version 2023.03.0) [40], and general power analysis procedures [41,42]. To examine the influence of ethnicity on the relationship between genotypes and PTSD risk, all meta-analytic models were re-run with ethnicity included as a factor (meta_analysis **←** ma (yi = effect_size, vi = variance, mods = ~ethnicity, data = data, method = “REML”). In detail, mods = ~ ethnicity specifies that ethnicity is considered as a moderator (factor) in the meta-analysis.

As we used only previously available data, we did not consider it necessary to obtain ethical approval or written informed consent. Statistical analyses were performed using JASP software (JASP Team, 2022). JASP (version 0.16.1) was used together with the software for meta-analyses.

## 3. Results

### 3.1. Study Selection

The study selection process is summarized in the flow diagram (Figure 1) leading to 17 studies [26,27,28,29,32,43,44,45,46,47,48,49,50,51,52] that were included in the systematic review (Table 1). From a total of 93 potentially eligible studies, after duplicate removal, 14 studies were excluded because they were not focused on 5HTTLPR; the other 14 studies were excluded because of no direct association between PTSD and 5HTTLPR, and seven studies were excluded because of the absence of a control group. The characteristics of studies eligible for inclusion are described in Table 1, including year of publication, study design, number of PTSD cases, sample of controls, number and percentage of males, mean age, diagnostic instrument used to assess PTSD, type of PTSD assessed, and biallelic or triallelic genotype approach.

Fifteen studies described a biallelic analysis (fourteen [26,27,29,32,43,45,46,47,48,49,50,51,52] provided the frequencies for the biallelic frequency model (BFM), and all fifteen studies [26,27,29,32,43,45,46,47,48,49,50,51,52,53] provided frequencies for the biallelic dominant model (BDM); therefore, they could be included in the meta-analysis of the biallelic approach; seven studies reported a triallelic analysis (seven studies [26,28,44,45,48,49,51] provided the frequencies for the triallelic frequency model (TFM), and seven studies provided frequencies for the triallelic dominant model (TDM)) [26,28,44,45,48,49,51] and consequently they could be inserted into the meta-analysis of the triallelic approach (see Table 2). In the biallelic meta-analysis, there were 2586 PTSD patients and 8838 controls, while in the triallelic meta-analysis, there were 627 and 3524, respectively.

### 3.2. Meta-Analysis of the Allelic Association with PTSD

#### 3.2.1. Biallelic Approach

The association between genotypes and the risk of PTSD was not significant when examining the allele frequency (S vs. L) in the biallelic approach and was low and statistically not significant (LogOR = 0.098; 95% CI [−0.139, 0.335]; *p* = 0.417; ω^2^ = 0.099). True heterogeneity across studies was large (I2 = 84.011%; Q = 49.281; df = 13; *p* < 0.001). The funnel plot showed slight asymmetry. The trim-and-fill method suggests that one additional study would be required to make the plot symmetric on the right side (Figure 2A).

The dominant model, (BDM, S+ vs. LL), gave similar findings showing a low and statistically not significant association between genotypes and the risk of PTSD (LogOR = 0.037; 95% CI [−0.245, 0.320]; *p* = 0.795; ω^2^ = 0.046). True heterogeneity across studies was large (I2 = 77.150%; Q = 41.980; df = 15; *p* < 0.001). Visual inspection of the funnel plot highlighted slight asymmetry on the right side. The trim-and-fill method revealed that one further study is needed to make the plot symmetric (Figure 2B).

In supplementary analyses, we inserted race in order to test the influence of race on the tested association models; however, no significant impact emerged in either the biallelic frequency or dominant models (all *p*s > 0.05).

#### 3.2.2. Triallelic Approach

The association between genotypes and the risk of PTSD was not significant even in the triallelic approach, more precisely in the triallelic frequency model (TFM, S’ vs. L’) (LogOR = 0.118; 95% CI [−0.185, 0.420]; *p* = 0.446; ω^2^ = 0.074). True heterogeneity across studies was moderate (I2 = 72.214%; Q = 22.236; df = 6; *p* = 0.001). The funnel plot showed an overall symmetric distribution of studies, and the trim-and-fill procedure revealed that no further study has to be added to make the plot symmetric (Figure 3A).

The triallelic dominant model (TDM, S’S’ + S’L’ vs. L’L’) confirmed the lack of a significant association between genotypes and the risk of PTSD (LogOR = 0.015; 95% CI [−0.328, 0.357]; *p* = 0.934; ω^2^ = 0.027). True heterogeneity across studies was low (I2 = 44.737%; Q = 10.522; df = 6; *p* = 0.104). The funnel plot was symmetric, and the trim-and-fill procedure proposes that no further studies would be necessary to make the plot symmetric (Figure 3B).

Finally, additional analyses including the race in meta-analytic models revealed a significant impact of race only in the triallelic dominant model (TDM) with a positive association between genotypes and the risk of PTSD (race Africans: z = 2.118, *p* = 0.034); no significant impact of race emerged in the triallelic frequency model (TFM) (*p* > 0.05).

## 4. Discussion

This is the first meta-analysis we are aware of that explicitly examines how ethnicity affects the association between 5-HTTLPR and PTSD. Ethnicity should be included in the meta-analysis as previous analyses [36,37,38] indicate heterogeneity between studies, with the exception of analyses for highly traumatized individuals and ethnic groups, which showed no heterogeneity. We employed several strategies to address this issue, such as meta-regression to examine the effects of the percentage of different ethnic groups (e.g., European/Caucasian or African American/African American) and subgroup analyses to assess ethnicity. Our results suggest that the association between 5-HTTLPR and PTSD is significantly influenced by ethnicity, particularly African descent. For example, ref. [54] found racial differences between Caucasians and Asians in the frequency of 5-HTTLPR alleles. In their study of healthy Caucasian individuals from Croatia and Russia, significant differences in genotype and allele frequencies were discovered, particularly in the frequency of the S allele and the S/S genotypes. These racial differences may lead to contradictory results in correlational studies looking at behavior, personality traits, and mental illness in different racial groups. Similarly, ref. [55] documented remarkable variation in allele frequencies and revealed the presence of ultra-long (“XL”) 5-HTTLPR alleles in African populations that were absent in Native American and Caucasian populations.

These differences highlight the importance of accurately classifying L-G alleles to prevent genotype misclassification and to investigate how XL alleles affect the transcriptional efficiency of SLC6A4 [56]. Although these results are consistent with previous meta-analyses [36,37], no significant correlation between PTSD and triallelic or biallelic SNPs was found in our data. A plausible conclusion is that there may not be a direct correlation between 5-HTTLPR and PTSD. However, given the growing body of research linking 5-HTTLPR polymorphisms to a variety of stress-related psychopathologies [19], a number of alternative theories need to be considered.

The focus of studies in this area has been on a third functional allele present in 5-HTTLPR polymorphisms. While there are at least ten known allelic variations of 5-HTTLPR, our data do not support significant differences between biallelic and triallelic studies. Nevertheless, it is not yet known whether this variability influences the strength of the association. Intriguingly, a meta-analysis found that women were more likely to suffer from PTSD following trauma than men [57], suggesting that there may be gender differences in susceptibility to PTSD that need to be further explored. Secondly, we observed low post hoc statistical power in our analyses. Problems with statistical power could arise from several factors and may explain the lack of a meaningful correlation. In the case of highly polygenic disorders, such as PTSD, the effect size of a particular allele is likely to be small, as seen in extensive genetic studies of other mental disorders. Additionally, our meta-analysis encountered significant variability in sample sizes across the included studies. The presence of smaller sample sizes in some studies may have contributed to underpowered analyses. Given the expected small magnitude of effects and the diversity in research approaches and PTSD subtypes investigated, it is conceivable that the statistical power was insufficient to detect a meaningful correlation. To improve the precision of these estimates, more extensive association studies and meta-analyses that include larger sample sizes are necessary. Finally, 5-HTTLPR may not have a direct effect on PTSD. Rather, it may be involved in gene-environment (GxE) interactions in which the genotype mitigates the effects of environmental stressors [40,58]. The findings are consistent with the differential susceptibility hypothesis, which posits that variations in 5-HTTLPR may indicate broader susceptibility to environmental exposure, with differential effects depending on the type of exposure [59]. The limited number of studies that have looked at GxE interactions in this setting, as well as the wide variety of stressors studied, make it difficult to conduct formal meta-analyses of these interactions at present. Further research is needed to learn more about these complicated relationships and the potential importance of specific genes in PTSD [37].

## 5. Conclusions

When analyzing our data, it is crucial to take into account a few significant constraints, including the relatively small sample size of included studies, poorer quality features, and moderate to substantial levels of variability. Furthermore, even if the chosen studies included a control group, special attention should be given to factors that might compromise the internal validity of the main observational studies, such as the various research designs, the kind of PTSD that was evaluated, and the presence of psychiatric comorbidities. Using two potential approaches (allele frequency and dominant models), our systematic review and meta-analysis, despite these limitations, do not suggest a direct main impact of 5-HTTLPR polymorphisms on PTSD. It is interesting to note that there are some early data suggesting that ethnicity may be important in modulating the association between 5-HTTLPR and PTSD and they should be taken into account in future studies. Furthermore, further research on GxE interactions and epigenetic modification may reveal other environmental factors that might interact with the PTSD-related 5-HTTLPR mutations.

## Figures and Tables

**Figure 1 genes-15-01270-f001:**
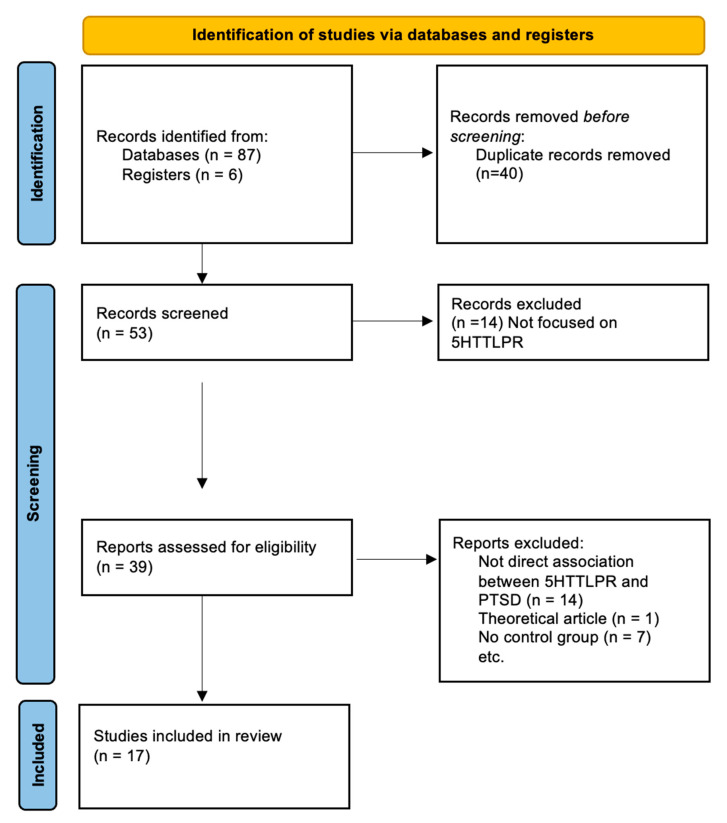
PRISMA flow diagram.

**Figure 2 genes-15-01270-f002:**
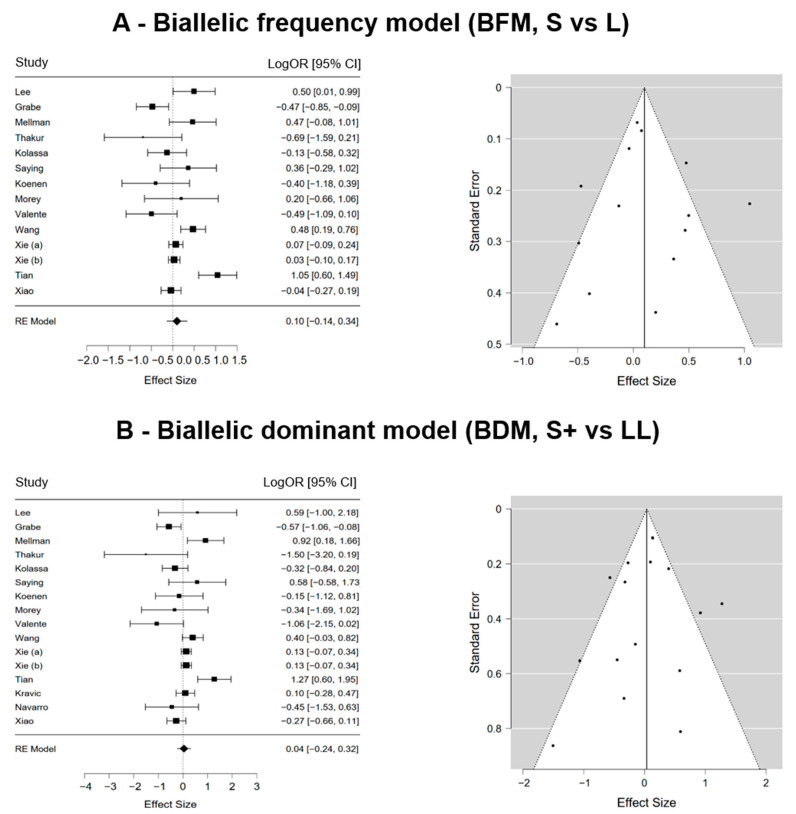
Forest plots and funnel plots of the 5-HTTLPR biallelic frequency model (S vs. L, panel (**A**)) or 5-HTTLPR biallelic dominant model (S+ vs. LL, panel (**B**)) and post-traumatic stress disorder (PTSD).

**Figure 3 genes-15-01270-f003:**
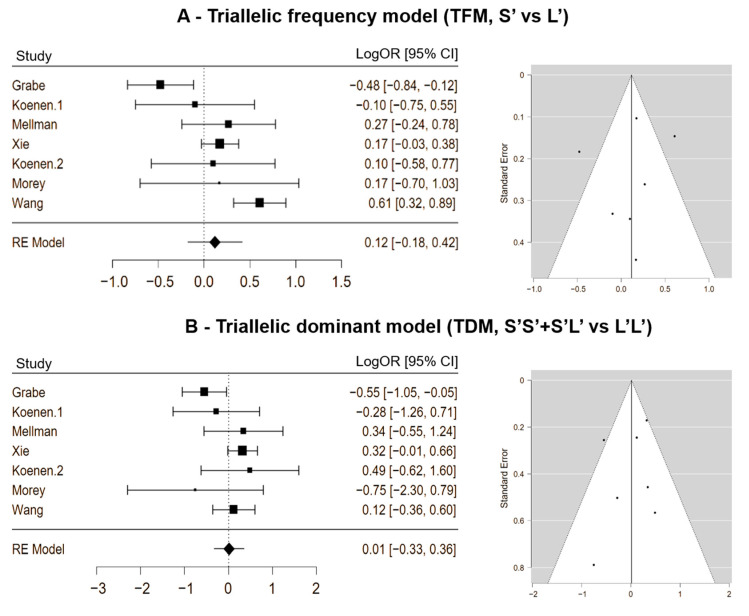
Forest plots and funnel plots of the 5-HTTLPR triallelic frequency model (S’ vs. L’, panel (**A**)) or 5-HTTLPR triallelic dominant model (S’S’ + S’L’ vs. L’L’, panel (**B**)) and post-traumatic stress disorder (PTSD).

**Table 1 genes-15-01270-t001:** Characteristics of the study.

Ref.	Year	PTSD N	Controls N	No. of Participants	Age, Range or Means	Ethnicity	Study Design	Diagnostic Instrument	PTSD Measure	Allele Sensitivity	Results
[43]	2005	100	197	297	35.29	Asian	Cross-sectional	SCID	Last 9 months	Biallelic	Compared to normal controls, PTSD patients had a considerably greater frequency of the s/s genotype. These results imply that one of the genetic variables contributing to vulnerability to PTSD is the SERTPR s/s genotype.
[26]	2009	67	1596	3045	20–79	European	Longitudinal	SCID	Lifetime	Triallelic	An additive excess risk in the LA/LA group for frequent trauma
[44]	2009	19	571	590	<60 (22.7%)	Mixed	Cross-sectional	National Women’s Study PTSD Module	Last 6 months	Triallelic	According to analysis, the S-allele was linked to a higher risk of PTSD in high-risk situations but a decreased risk of PTSD in low-risk circumstances (low crime/unemployment rates).
[45]	2009	55	63	118	39.9	African Americans	Case–Control	CAPS SCID	Lifetime	Triallelic	No significant association between 5HTTLPR and PTSD
[28]	2009	229	1023	1252	38.97	Mixed	Cross-Sectional	SSADDA	Lifetime	Triallelic	While the 5-HTTLPR gene by itself was not sufficient to predict the development of PTSD, interaction between adult traumatic experiences and childhood adversity enhanced the likelihood of PTSD, especially for individuals with high rates of exposure to both forms of trauma.
[46]	2009	24	17	41	31.96	Mixed	Cohort study	CAPS	Last year	Biallelic	At 12 months, the LL homozygotes found a higher chronic PTSD rate compared to those with SS and SL genotypes.
[27]	2010	331	77	408	17–68	African	Cross- sectional	Post-traumatic Diagnostic Scale	Lifetime	Biallelic	PTSD was most likely to develop in those who were homozygous for the short allele of the SLC6A4 promoter polymorphism.
[47]	2010	29	48	77		Turkish	Cohort study	CAPS	Lifetime	Biallelic	While the development of PTSD was not affected by any of the 5-HTT polymorphisms examined, people who have experienced PTSD may have fewer hyperarousal symptoms if they carry the L-allele for a 5-HTT gene-related polymorphic area.
[48]	2011	23	77	100	45.32	African Americans	Cross-sectional	Modified PTSD Checklist	Lifetime	Triallelic	The interaction between the number of traumatic events and the 5HTTLPR genotype was tested, but it was not significant.
[49]	2011	22	20	42	30.8	Caucasian	Cross-sectional	Davidson Trauma scaleMRI scanner	Current	Biallelic	SLC6A4 SNP rs16965628 and 5-HTTLPR are associated with a bias in neural responses to traumatic reminders and cognitive control of emotions in patients with PTSD.
[50]	2011	65	34	99	39.99	Brazilians	Case–Control	SCID-I and CAPS	Current	Biallelic	No association between 5HTTLPR and PTSD.
[51]	2011	212	176	388	49.06	Mixed	Case–Control	CAPS, MINI, CES	Current	Triallelic	After experiencing trauma, the low transcriptionally efficient 5-HTTLPR genotype SS increases the likelihood of developing PTSD.
[29]	2012	AA321	2078	2399	42.2	Mixed	Cross-Sectional	SSADDA	Lifetime	Biallelic	Those who experienced hardship as children are more likely to develop post-traumatic stress disorder (PTSD) if they have one or two copies of the 5-HTTLPR S gene.
		EA 398	2381	2779	39	Mixed	Cross-Sectional	SSADDA	Lifetime	Biallelic	Those who experienced hardship as children are more likely to develop post-traumatic stress disorder (PTSD) if they have one or two copies of the 5-HTTLPR S gene.
[32]	2015	64	119	183	15.2	Asian	Cross-sectional	PCL-C	Last 3 years	Biallelic	Polymorphisms of 5-HTTLPR and exposure to earthquakes had statistically significant positive effects on PTSD. The interaction exposure to earthquakes and 5-HTTLPR polymorphisms both demonstrated statistically significant favorable impacts on PTSD. The 5-HTTLPR/earthquake exposure interaction effects were statistically significant.
[52]	2019	287	280	565	15.0	Tibetan	Cross-sectional	PCL-C SCID	Current	Biallelic	No significant association between 5HTTLPR and PTSD.

**Table 2 genes-15-01270-t002:** Frequencies of the study.

Ref.	Year	PTSD					CONTROL					PTSD					CONTROL				
		S	L	SS	SL	LL	S	L	SS	SL	LL	S’	L’	S’S’	S’L’	L’L’	S’	L’	S’S’	S’L’	L’L’
[43]	2005	175	25	77	21	2	319	75	129	61	7	-	-	-	-	-	-	-	-	-	-
[26]	2009	40	94	6	28	33	1294	1898	264	766	566	48	86	8	32	27	1512	1680	364	784	448
[44]	2009	-	-	-	-	-	-	-	-	-	-	17	21	4	9	6	539	603	116	307	148
[45]	2009	42	70	5	32	19	35	93	7	21	36	66	46	20	26	10	67	61	18	31	15
[28]	2009	-	-	-	-	-	-	-	-	-	-	234	224	58	118	53	957	1089	234	489	300
[46]	2009	23	25	8	7	9	22	12	7	8	2	-	-	-	-	-	-	-	-	-	-
[27]	2010	112	550	15	82	234	29	125	1	27	49	-	-	-	-	-	-	-	-	-	-
[47]	2010	30	28	6	18	5	41	55	6	29	13	-	-	-	-	-	-	-	-	-	-
[48]	2011	10	34	1	8	13	45	103	12	21	41	21	23	4	13	5	67	81	17	33	24
[49]	2011	22	22	7	8	7	18	22	3	12	5	26	18	10	6	6	22	18	5	12	3
[50]	2011	55	71	13	29	21	38	30	9	20	5	-	-	-	-	-	-	-	-	-	-
[51]	2011	207	217	56	95	61	131	221	21	89	66	261	163	94	73	45	164	188	29	106	41
[29]	2012	222	650	26	170	240	1269	3997	170	929	1534	-	-	-	-	-	-	-	-	-	-
[29]	2012	460	564	96	268	148	2511	3187	566	1379	904	-	-	-	-	-	-	-	-	-	-
[32]	2015	79	46	31	14	16	86	151	33	19	66	-	-	-	-	-	-	-	-	-	-
[52]	2019	302	272	93	116	78	300	260	82	136	62	-	-	-	-	-	-	-	-	-	-

## Data Availability

No new data were created or analyzed in this study.

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
