# Peer review of "The Association between Post-Traumatic Stress Disorder, 5HTTLPR, and the Role of Ethnicity: A Meta-Analysis"

_genes, 2024, doi:10.3390/genes15101270_

Round 1

Reviewer 1 Report

Comments and Suggestions for Authors

I have several comments to the authors outlined below

1. What does LG stand for? The Nakamura et al. manuscript reports several alleles with different repeat lengths. So, the question is, what length is the LG allele the authors are referring to here?

2. The authors need to clarify their exclusion criteria. For instance, they say 14 studies were excluded because of no direct association between PTSD and HTTLRP. What does that mean exactly? Do they mean that HTTLRP was not associated with PTSD, or that 5HTTLRP was not tested for an association with PTSD? If the 14 studies reported no association between PTSD and HTTLRP, why were they excluded? 

3. As a potential reason for the lack of statistically significant findings in their meta-analysis, the authors state a lack of sufficient power. However, it is unclear whether that is a general statement that the existing studies are underpowered or whether they refer to their meta-analysis. For their study, at least, they could have done the power analysis to confirm or rejects that assertion. 

Comments on the Quality of English Language

none

Author Response

Dear Editor,

We revised the manuscript entitled The Association Between Post Traumatic Stress Disorder “ , submitted to the Genes Journal.

We thank the editor and the reviewers for taking the time for reviewing our article. 

We are grateful for the meaningful suggestion they gave us.

We have followed the reviewers’ suggestions, trying to answer all comments.

You can find our response for each point highlight below.

Reviewer #1

Comment#1: What does LG stand for? The Nakamura et al. manuscript reports several alleles with different repeat lengths. So, the question is, what length is the LG allele the authors are referring to here?

Reply#1: We thank the reviewer for the comment. It is a different way of classification. According to the study, an A/G nucleotide substitution in the L allele (rs25531) renders the 5-HTTLPR triallelic, with the LG allele functionally equivalent to the S allele compared to the LA allele in vitro

Comment#2: The authors need to clarify their exclusion criteria. For instance, they say 14 studies were excluded because of no direct association between PTSD and HTTLRP. What does that mean exactly? Do they mean that 5HTTLRP was not associated with PTSD, or that 5HTTLRP was not tested for an association with PTSD? If the 14 studies reported no association between PTSD and HTTLRP, why were they excluded? 

Reply#2: It means that 5HTTLPR was not studied for a direct association with PTSD

Comment#3: As a potential reason for the lack of statistically significant findings in their meta-analysis, the authors state a lack of sufficient power. However, it is unclear whether that is a general statement that the existing studies are underpowered or whether they refer to their meta-analysis. For their study, at least, they could have done the power analysis to confirm or rejects that assertion. 

Reply#3: Our meta-analysis encountered notable variability in sample sizes among the included studies, with participant counts ranging from 41 to 3045. The presence of small sample sizes in some studies may have resulted in underpowered analyses, potentially leading to non-significant findings due to insufficient statistical power. This variability could impact the overall power and reliability of our meta-analytic conclusions (Griffin, J. W. (2021). Calculating statistical power for meta-analysis using metapower. The Quantitative Methods for Psychology). We added this issue in the Discussion section.

Reviewer 2 Report

Comments and Suggestions for Authors

Dear authors 

This is a very interesting meta-analysis. However, you have to pay attention to some points. 

Clarify the title without acronyms.

The abstract should provide the main results of the research.

In the introduction, you start wrong with the definition of PTSD. According to DSM 5, criteria levels are 4, not 3. Please correct and add citations. 

The purpose and necessity of the research are not shown in the introduction. Please specify at the end of the section. 

Did you only use these key terms in the search? Why? 

You do not specify whether you have registered with PROSPERO. 

The methodological quality of the articles is lacking.

The discussion is overanalyzed. You could collapse it. 

Comments on the Quality of English Language

Minor

Author Response

Dear Editor,

We revised the manuscript entitled The Association Between Post Traumatic Stress Disorder “ , submitted to the Genes Journal.

We thank the editor and the reviewers for taking the time for reviewing our article. 

We are grateful for the meaningful suggestion they gave us.

We have followed the reviewers’ suggestions, trying to answer all comments.

You can find our response for each point highlight below.

Comment#1: Clarify the title without acronyms.

The abstract should provide the main results of the research.

Reply#1: We thank the reviewer for the comment, and we changed accordingly.

Comment#2: In the introduction, you start wrong with the definition of PTSD. According to DSM 5, criteria levels are 4, not 3. Please correct and add citations. 

The purpose and necessity of the research are not shown in the introduction. Please specify at the end of the section. 

Reply#2: We added in the introduction

Comment#3:  

  • Did you only use these key terms in the search? Why? 
  • You do not specify whether you have registered with PROSPERO. 
  • The methodological quality of the articles is lacking.

Reply#3:

  • Yes, we used those key terms as summarized key words in all the research of the topic. We manually looked for the ethnicity factor.
  • We did not register the meta-analysis with PROSPERO
  • We have summarized the main features of the articles

Comment#4: The discussion is overanalyzed. You could collapse it. 

Reply#4: We have followed the reviewer’s advice

Round 2

Reviewer 1 Report

Comments and Suggestions for Authors

A few additional comments on the revised manuscript:

1. I still do not see the power analyses of their meta-analysis. Presenting power would allow the potential reader to gauge better the validity of the author's results. 

2. The revised manuscript is missing important information (that I have not noticed in my previous review) about the meta-analysis. Specifically, more detailed information is needed on their approach to how they modeled the multi-ancestry analysis. Modeling multi-ancestry information in a meta-analytical framework is not trivial, and the authors need to provide greater details; how exactly have they done that? 

Comments on the Quality of English Language

English language is fine

Author Response

Dear reviewer,

thank you for the insight you have given us.

We have clarified your comments.

A few additional comments on the revised manuscript:

  1. I still do not see the power analyses of their meta-analysis. Presenting power would allow the potential reader to gauge better the validity of the author's results. 

Reply: In line with the reviewer’s suggestion, we have included post-hoc power analyses to provide insight into the sensitivity of our meta-analytic models. We calculated observed power for all meta-analytic models using resources available in the metafor package in R (Viechtbauer, W., 2010, Conducting Meta-Analyses in R with the metafor Package), along with general power analysis procedures described in Hedges and Pigott (2001, The Power of Statistical Tests in Meta-Analysis) and Borenstein et al. (2009, Introduction to Meta-Analysis).

While we have added post-hoc power calculations in the revised version of the manuscript (see Statistical Analyses and Resultssections), it is important to emphasize that the low observed power in our meta-analytic models may be misleading. This is because post-hoc power relies on the observed data, which may not accurately reflect the true likelihood of detecting an effect for several reasons:

  1. Small Sample Sizes in Individual Studies: The precision of effect size estimates is lower when the individual studies included in the meta-analysis have small sample sizes (our meta-analysis encountered notable variability in sample sizes). This leads to higher standard errors, thereby reducing the overall statistical power of the meta-analysis.
  2. Limited Number of Studies: When few studies are available, the meta-analysis has limited precision, making it difficult to draw strong conclusions about the overall effect size. This further contributes to low observed power.
  3. Heterogeneity and High Variance in Outcome Measurements: In our meta-analysis, outcome measures were highly variable due to differences in measurement scales, definitions, and timing across studies. This variability, or heterogeneity, reduces the consistency of effect sizes across studies and lowers the likelihood of detecting a significant combined effect, thereby reducing statistical power.

It is also important to note that post-hoc power is often criticized because it is directly tied to the p-value obtained in the analysis. Many statisticians argue that post-hoc power does not provide meaningful information beyond the p-value. Specifically, it is highly sensitive to the observed effect size and variance—when the effect size is close to zero, power will naturally be low.

We have addressed and discussed these potential aspects affecting post-hoc power in our meta-analyses in the Discussion section.

  1. The revised manuscript is missing important information (that I have not noticed in my previous review) about the meta-analysis. Specifically, more detailed information is needed on their approach to how they modeled the multi-ancestry analysis. Modeling multi-ancestry information in a meta-analytical framework is not trivial, and the authors need to provide greater details; how exactly have they done that? 

Reply: To explore the impact of ethnicity on the relationship between genotypes and PTSD risk, all meta-analytic models were re-run with ethnicity included as a factor (meta_analysis <- rma(yi = effect_size, vi = variance, mods = ~ ethnicity, data = data, method = "REML"). In details, mods = ~ ethnicity specifies that ethnicity is considered as a moderator (factor) in the meta-analysis.

Reviewer 2 Report

Comments and Suggestions for Authors

Dear Authors

You did a great job and you have addressed all the issues. 

Kind regards

Comments on the Quality of English Language

Minor

Author Response

Dear reviewer,

thank you for all your insights.
